# Long-Lasting Anti-Inflammatory and Antinociceptive Effects of Acute Ammonium Glycyrrhizinate Administration: Pharmacological, Biochemical, and Docking Studies

**DOI:** 10.3390/molecules24132453

**Published:** 2019-07-04

**Authors:** Francesco Maione, Paola Minosi, Amalia Di Giannuario, Federica Raucci, Maria Giovanna Chini, Simona De Vita, Giuseppe Bifulco, Nicola Mascolo, Stefano Pieretti

**Affiliations:** 1Department of Pharmacy, School of Medicine and Surgery, University of Naples Federico II, Via Domenico Montesano 49, 80131 Naples, Italy; 2National Centre for Drug Research and Evaluation, Istituto Superiore di Sanità, Viale Regina Elena 299, 00161 Rome, Italy; 3Department of Pharmacy, University of Salerno, Via Giovanni Paolo II 132, 84084 Fisciano (SA), Italy

**Keywords:** ammonium glycyrrhizinate, docking, long-lasting effect, nociception, inflammation

## Abstract

The object of the study was to estimate the long-lasting effects induced by ammonium glycyrrhizinate (AG) after a single administration in mice using animal models of pain and inflammation together with biochemical and docking studies. A single intraperitoneal injection of AG was able to produce anti-inflammatory effects in zymosan-induced paw edema and peritonitis. Moreover, in several animal models of pain, such as the writhing test, the formalin test, and hyperalgesia induced by zymosan, AG administered 24 h before the tests was able to induce a strong antinociceptive effect. Molecular docking studies revealed that AG possesses higher affinity for microsomal prostaglandin E synthase type-2 compared to type-1, whereas it seems to locate better in the binding pocket of cyclooxygenase (COX)-2 compared to COX-1. These results demonstrated that AG induced anti-inflammatory and antinociceptive effects until 24–48 h after a single administration thanks to its ability to bind the COX/mPGEs pathway. Taken together, all these findings highlight the potential use of AG for clinical treatment of pain and/or inflammatory-related diseases.

## 1. Introduction

The most prevalent medical issues strongly affecting people in terms of health and quality of life are acute and chronic pain [1]. Acute and chronic pain are different clinical entities. Acute pain resolves with healing of the underlying injury and it is usually nociceptive, whereas chronic pain is a pathophysiological state connected with the alteration of the peripheral and/or central nervous systems and it is commonly coupled with inflammation caused by tissue trauma, chemical stimuli, and infectious agents. A considerable number of studies have demonstrated that several mediators, including cytokines (Interleukin (IL) -1β, IL-6, IL-17, IL-10, tumor necrosis factor (TNF) -α), chemokines (chemokine (CXCL1), Motif Chemokine Ligand (CCL2), C-X-C chemokine receptor type 2 (CXCR2)), lipid mediators (prostaglandins and leukotrienes) and growth factors (nerve growth factor (NGF), and brain-derived neurotrophic factor (BDNF)), play an important role in inflammatory pain [2].

Two types of treatments can be used to treat inflammatory pain: nonsteroidal anti-inflammatory drugs (NSAIDs) and opioids. Very often there are several side effects, such as gastrointestinal lesions [3] and nephrotoxicity [4], in the case of treatment with NSAIDs, when they are used for a long time, and respiratory depression, tolerance, and physical dependence for opioids [5]. Thus, the identification of new potential targets to treat pain without inducing side effects is crucial.

*Glycyrrhiza glabra L*., commonly known as liquorice, is a perennial, herbaceous shrub, belonging to the family of Leguminosae. This plant is endemic to Mediterranean countries, such as Greece, Spain, and Southern Italy [6]. Liquorice root has been used since prehistoric times. It contains triterpenoid saponins (3–5%), mainly glycyrrhizin, a mix of calcium and potassium salts of 18β-glycyrrhizic acid (also known as glycyrrhizic or glycyrrhizinic acid and a glycoside of glycyrrhetinic acid) and flavonoids (1–1.5%) [7,8]. Responsible for the anti-inflammatory effect, owing an indirect strengthening of the glucocorticoid activity, are triterpenes [9]. Liquorice is a herb that people have used for thousands of years to treat a variety of ailments, such as dermatitis, psoriasis, and eczema, and shows comparable efficacy to that of corticosteroids [10,11,12]. In particular, the ammonium salt of glycyrrhizic acid (AG) has strong anti-inflammatory activity [13]. Furthermore it is clear that glycyrrhizin leads to a decrease in inflammatory events due to spinal cord injury (edema, tissue damage, apoptosis, inducible expression of nitric oxide synthase (iNOS)) and nuclear factor kappa-light-chain-enhancer of activated B cells (NFκB) activation, improving the recovery of limb function [14].

Antinociceptive effects were also reported for glycyrrhizic acid and derivatives. Disodium glycyrrhetinic acid hemiphthalate significantly suppressed acetic acid-induced writhing responses in mice, with a potency ~12 times higher than acetylsalicylic acid [15]. Glycyrrhizin in mice significantly inhibited the nociception induced by acetic acid and formalin, downregulating the expression levels of TNF-α, IL-6, iNOS, and COX-2 [16]. Recent evidence confirm the anti- inflammatory and antinociceptive effectiveness of glycyrrhizin and suggest that these effects depend upon the glycyrrhizin inhibition of microglial high-mobility group box 1 protein (HMGB1) [17].

The above-mentioned activities were observed in a short time range after administration of AG and AG derivates. Therefore, in the present study, we investigated the long-lasting anti-inflammatory and antinociceptive effects of AG after a single administration and we have explored its mechanism of action by biochemical and molecular docking studies.

## 2. Results

### 2.1. Edema Induced by Zymosan

In animals treated with vehicle (Hepes, 10 mL/kg, i.p.) 10 min before zymosan, we discovered a rise in paw volume that achieved the maximal value 3–4 h after the injection, followed by a slight reduction in the following 48 h (Figure 1). In this set of experiments, we noticed significant differences in the behavioral responses between treatments (F2,33 = 16.30, *p* < 0.0001) and in the time elapsed after zymosan administration (F5,165 = 38.59, *p* < 0.0001). The i.p. administration of AG at the dose of 50 mg/kg 10 min before zymosan generated a considerable reduction of paw edema induced by zymosan injection, from 1 to 3 h after zymosan injection (Figure 1). The i.p. administration of AG at the dose of 150 mg/kg induced a robust reduction of paw edema starting from 1 h and lasting for the full course of treatment (Figure 1).

### 2.2. Writhing Test

The antinociceptive effect of AG in acetic acid writhing test is shown in Figure 2. Statistical analysis revealed significant differences between treatments (F2,24 = 17.69, *p* < 0.0001). In this test, AG administered i.p. at the dose of 50 mg/kg reduced writhes induced by acetic acid. Severe inhibition of the number of writhes was discovered when AG was administered at the dose of 150 mg/kg.

### 2.3. Formalin Test

Subcutaneous injection of formalin induced a nociceptive behavioural response that showed a biphasic trend. There was an early phase (from 0 to 10 min after formalin injection) produced by the direct stimulation of peripheral nociceptors, and a late prolonged phase (from 15 to 40 min) which reflected the response to inflammatory pain. The total time the animal spent licking or biting its paw during the early and late phase of formalin-induced nociception was recorded. The results obtained in these experiments are reported in Figure 3. The administration of AG at the dose of 50 or 150 mg/kg i.p. 24 h before formalin, did not modify the nociceptive response induced by aldehyde in the early phase of the test (F2,27 = 2.903, *p* > 0.05). A considerable decrease of the formalin-induced licking and biting activity was instead observed in the late phase of the test (F2,27 = 24.69, *p <* 0.0001). When the confront was restricted to two means, AG administered at the dose of 50 mg/kg induced a light but nonsignificant reduction of formalin-induced behaviour (*p* > 0.05) in the late phase. On the contrary, AG administered at the dose of 150 mg/kg strongly reduced the nociceptive behavior induced by formalin (*p* < 0.0001).

### 2.4. Zymosan-Induced Hyperalgesia

This experimental pain model is characterized by the measurements of time-dependent hyperalgesia after zymosan administration. This measurement is equivalent to time-dependent reduction in the latency to respond to the thermal stimuli applied to the injected paw compared with the baseline measurements. In particular, 20 μL of zymosan A (2.5% *w/v* in saline) was administered s.c. into the dorsal surface of one hind paw.

In our experiments, treatments were given i.p. 10 min before the first measurement of the pain threshold, i.e., 1 h after zymosan administration. Figure 4 shows the results of these experiments. With the use of two-way ANOVA, we can notice that there are significant differences in treatments (F2,27 = 6.901, *p* < 0.01) and in the time point when pain threshold was recorded (F5,135 = 30.51, *p* < 0.0001). Tukey’s multiple comparison test showed significant differences from 1 to 24 h after zymosan administration between animals treated with AG at the dose of 150 mg/kg, i.p. and those deal with vehicle. AG administered at low dose (50 mg/kg, i.p) induced a light but nonsignificant increase in pain threshold (Figure 4).

### 2.5. Zymosan Peritonitis and Cytokines-Chemokines Protein Array

To investigate the molecular mechanisms behind the selective long-lasting anti-inflammatory and antinociceptive effects of AG, we firstly compared the recruitment of total inflammatory cells and the cytokines and chemokines profiles of collected inflammatory fluids obtained from mice receiving zymosan and zymosan plus AG at 4 and 24 h. As shown in Figure 5a, 4 h after zymosan (500 mg/kg) or zymosan + AG (150 mg/kg) injection, mice exhibited no significant difference in the number of recruited inflammatory leukocytes. In contrast, at 24 h, mice receiving AG (150 mg/kg) showed a marked decrease (*p* < 0.01) in the number of inflammatory infiltrates compared to zymosan-treated mice (Figure 5b). Administration of AG at 50 and 150 mg/kg did not show any significant effect, both at 4 and 24 h (data not shown). Contingently, as shown in Figure 6, zymosan administration at 4 h caused a massive production of cyto/chemokines (Figure 6a) that was not counterbalanced in AG-treated mice (Figure 6b), except for macrophage inflammatory proteins (MIP)-1α and keratinocyte chemoattractant (KC) production and with a less extent IL-6, SDF-1 and TNF-α (Figure 6c). Interestingly, comparison of the inflammatory profile of fluids from zymosan (Figure 7a) and zymosan + AG (Figure 7b) at 24 h revealed a striking difference in the semiquantitative levels of IL-6, IL-10, IL-1β, IP-10, KC, MCP-5, macrophage inflammatory proteins (MIP) 1-α, 1-β, 2, and TNF-α (Figure 7c,d).

### 2.6. Docking Experiments

The molecular docking studies were used to evaluate and to rationalize the anti-inflammatory activity of glycyrrhizic acid towards some key enzymes of the arachidonic acid cascade such as 5-lipoxygenase (5-LO), 5-lipoxygenase-activating protein (Appendix A), microsomal prostaglandin E synthase 1 and 2 (mPGES-1 and mPGES-2), and COX-1 and COX-2, responsible for leukotrienes (LTs) and prostaglandin (PG) biosynthesis, respectively. These enzymes are involved in the generation of inflammatory mediators and their role in the progression of inflammation is well-established; therefore, they were suitable targets for the study of the glycyrrhizic acid anti-inflammatory activity.

### 2.7. mPGES-1 and mPGES-2

Prostaglandin E_2_ (PGE_2_) is one of the key mediators of inflammation and it is synthesized by members of the prostaglandin E synthase (PGESs) family from the unstable prostaglandin H_2_ (PGH_2_). The most known and studied isoforms are mPGES-1 and mPGES-2; the first one is an inducible isoform, whereas mPGES-2 is constitutively expressed [18]. They differ also in tissue localization, with mPGES-1 mainly expressed in lung, kidney, and reproductive organs and mPGES-2 mainly expressed in heart and brain [18,19]. Key amino acids for the binding are: Arg38, Asp49, Arg52, His53, Arg70, Arg73, Asn74, Glu77, Arg110, Leu121, Arg122, Arg126, Ser127, Ile125, Thr129, and Tyr130 for mPGES-1 [20,21,22,23], and Cys110, His241, His244, Ser247, Arg292, and Arg296 for mPGES-2 [24]. Molecular docking experiments involving both microsomal prostaglandin E synthase proteins revealed that AG has a higher affinity for mPGES-2 than mPGES-1. In particular, in the case of mPGES-1, the only interaction shown is one H-bond between the OH group and Arg52 and His53 on chain B. Regarding mPGES-2, instead, the three polar groups of AG are all involved in H-bonds with the target. In more detail:The OH group interacts both with the side chain and the backbone of Thr109,The carbonyl group interacts with Ser247, andThe carboxyl group forms an H-bond and a salt bridge with Arg296 (Figure 8 and Figure 9).

### 2.8. COX-1 and COX-2

Cyclooxygenases are the starting point in the production of eicosanoids, as they convert fatty acids into PGH_2_. Like mPGESs, they are present in two main isoforms: COX-1, which is constitutively expressed, and COX-2, which is inducible [25,26]. COX-1 and COX-2 monomers consist of three different domains: the EGF domain, the membrane-binding domain, and the catalytic domain containing the catalytic triad Arg120, Tyr355, and Glu524. Among these, the catalytic domain represents the main target for NSAID and, particularly, Arg120, Tyr355, Tyr385, Ser530, and Arg513 are essential for the inhibitory activity [27]. The study of the interaction between AG and the two isoforms of COX required a flexible docking approach (induced fit) [28] to allow the entrance of the ligand into the catalytic site of the enzymes. AG seems to locate better in the binding pocket of COX-2 as it interacts with key amino acids such as Trp387 and Ser530 (H-bonds) and Arg120 (salt bridge). In the case of COX-1, on the other hand, AG forms H-bonds and a salt bridge with Arg120 that do not reach the inner part of the binding pocket (Figure 10 and Figure 11). These data suggest a preference of glycyrrhetic acid (AG) for isoform 2 of COX.

### 2.9. 5-LO

The mammalian 5-lipoxygenase (5-LO) pathway produces potent mediators, such as leukotriene B4 (LTB4) and peptide leukotrienes (LTC4, LTD4 or LTE4), that have well-known functions and are involved in different pathologies and disorders [29,30,31]. To date, four different class of 5-LO direct inhibitors are classified: 1) iron-ligand inhibitors, 2) non-redox-active 5-LO inhibitors, which compete with arachidonic acid for binding to the enzyme 3) redox-active 5-LO inhibitors, interfering with the catalytic cycle of the enzyme, and 4) inhibitors that act with an unrecognized mechanism [30]. To computationally evaluate the binding of AG to 5-LO, a combined approach of rigid and flexible docking was used to predict its possible mechanism of action.

The human 5-LO structure (PDB ID: 3V99) [32] in complex with its substrate, arachidonic acid (AA), was used as the target for molecular docking studies [33]. The computational analysis highlighted two possible binding modes for the AG molecule (Figure 12), compatible with two different class of direct 5-LO inhibitors [30]. The results of the rigid docking approach are compatible with a non-redox type binding mode (Figure 12a), where AG was able to partially occupy the same binding site of AA and establishes hydrogen bonds with the side chain of Lys409 and with the backbone of Phe177. On the other hand, considering the induced fit molecular docking approach, the carboxylate group of AG was able to coordinate Fe^2+^ (Figure 12b) in a bidentate manner, and the OH group formed a hydrogen bond with the backbone of Arg666, behaving as an iron-ligand inhibitor.

These computational data have suggested a probable binding between AG and 5-LO by one of the two proposed mechanisms, which could represent a novel finding regarding its biological activity profile, especially with respect to glycyrrhizin from *Glycyrrhiza glabra L*. as reported by Chandrasekaran et al. [34].

## 3. Discussion

Available therapy for the management of chronic and inflammatory pain is not fully adequate in most cases. Furthermore, the common treatments of inflammatory pain, NSAIDs and opioids, lead to severe toxicities, including gastrointestinal lesions and nephrotoxicity [4] in the case of NSAIDs and respiratory depression, tolerance, and physical dependence for opioids when they are used for a long time [35]. Thus, identification of new potential targets which may affect pain and inflammatory processes is becoming an urgent clinical and therapeutic need. In our present study, we investigated the possible long-lasting anti-inflammatory and antinociceptive effects of AG using pharmacological, biochemical, and in silico tools. Our experimental method demonstrated that AG caused significant long-lasting reduction of inflammation and nociception, revealing that effects are most likely due to the inhibition of different aspects and mediators of inflammation. Moreover, these effects were observed at AG doses which have already been demonstrated to be free of toxic effects [36].

Several investigators have reported that the anti-inflammatory mechanism of AG involves cytokines such as IFN-γ, TNF-α, IL-1β, IL-4, IL-5, IL-6, IL-8, IL-10, IL-12, and IL-17 [37,38,39,40,41,42], intercellular cell adhesion molecule 1 (ICAM-1) and P-selectin, enzymes such as iNOS, and transcription factors such as NF-kappaB, signal transducer and activator of transcription (STAT)-3 and STAT-6 [33].

The anti-inflammatory efficacy of AG was evaluated using zymosan-induced paw edema and peritonitis. Zymosan consists of insoluble polysaccharides from yeast cell wall that induces an inflammatory reaction [43]. In this context, zymosan-induced ear or paw edema are preliminary models for screening potential anti-inflammatory drugs, characterized by pronounced dose- and time-dependent edema with moderate to severe infiltration of neutrophils and, to a lesser extent, macrophages and lymphocytes [44]. Furthermore, zymosan-induced paw edema is COX-2-dependent since it is significantly reduced in COX-2^−/−^ mice [45]. In this context, previous studies demonstrated that AG was able to reduce paw [15,16] or ear [16] edema formation and these effects were observed from 1 to 5 h after its administration [46]. After carrageenan injection in mouse paw, TNF-α, IL-6, iNOS, and COX-2 mRNA expressions increased and AG treatment (150 mg/kg, i.p.) markedly suppressed carrageenan’s effects [16]. The results of our experiments demonstrated that AG had a significant long-lasting anti-inflammatory activity since AG reduced paw edema formation until 48 h after administration.

Accordingly, to better investigate the long-lasting anti-inflammatory effect of AG, we next aimed to assess its action using zymosan-induced peritonitis. Intraperitoneal injection of zymosan is a model of acute inflammation and has been widely used for the quantification of cell types and inflammation-related soluble factors [47]. It is well known that the use of zymosan as experimental model of inflammation results in a range of benefits: following the injection with zymosan, it is possible to collect a reasonable amount of exudate for the analysis of several inflammatory mediators; the normal inflammatory response of an immunocompetent individual is imitated by zymosan injection. Furthermore, injection into a serosal cavity instead of an artificially formed cavity, such as a sterile air pouch, means that leukocytes exit from the site of inflammation via their natural conduits to the draining lymph node [48,49]. This model is simple, accurate, and capable of being reproduced [47].

When administered in the mouse peritoneal cavity, zymosan-induced acute inflammation characterized by increased vascular permeability, leukocyte influx, and release of inflammatory mediators [50]. In mice receiving AG 24 h before zymosan, AG induced a high decrease in the number of inflammatory cells and a significant decrease in IL-6, IL-10, IL-1β, IP-10, KC, MCP-5, MIP1-α, MIP1-β, MIP2, and TNF-α release in inflammatory fluids from zymosan-treated animals. This scenario was partially evoked even at 4 h post-AG administration. At this time-point, AG was able to reduce the levels of IL-6, MIP-1α, SDF-1, TNF-α, and KC. No data have previously been reported, to our knowledge, on AG effects in a zymosan-peritonitis model. However, in a model of acute lung injury induced by intratracheal administration of lipopolysaccharide (LPS) from *Escherichia coli*, AG was able to reduce lung edema, total leukocyte number, and neutrophil percentage in the bronchoalveolar lavage fluid (BALF), and AG downregulated TNF-α level in the lung [51]. Further investigation using a similar model of lung inflammation indicated that AG was able to reduce the concentrations of pro-inflammatory cytokines IL-1β in BALF and AG suppressed the expression of COX-2 and iNOS in lung tissue [52]. All reported data confirm the anti-inflammatory effects of AG, and these AG-related anti-inflammatory effects were until present 24–48 h after a single AG administration.

The writhing test, formalin test, and zymosan-induced hyperalgesia test were performed 24 h after AG administration to evaluate the possible long-lasting antinociceptive property of AG. Several studies have already demonstrated that AG is able to induce antinociceptive effects in the writhing test [15,16], formalin test, and thermal hyperalgesia [16] and these effects were observed from 1 to 5 h after AG administration. Our data demonstrated for the first time that AG exerts long-lasting antinociceptive effects, since AG-induced antinociception was observed until 24–48 h after a single administration.

To better investigate its analgesic profile, we finally tested the AG pharmacological profile in an acetic acid-induced writhing test, a commonly used method for monitoring preliminary antinociceptive activity, since in this test both central and peripheral analgesics are detected [53]. The injection of acetic acid directly activates the visceral and somatic nociceptors that innervate the peritoneum and induces inflammation not only in subdiaphragmatic visceral organs, but also in the subcutaneous muscle walls [54]. Furthermore, the nociceptive activity of acetic acid in the writhing model may be due to the release of TNF-α, IL-1β, and IL-8 by resident peritoneal macrophages and mast cells [55]. Our results revealed that AG administered 24 h before the test significantly reduced the number of writhes, suggesting that AG possessed long-lasting antinociceptive effects. Because of the lack of drug specificity, caution is required in interpreting the results obtained in the writhing test [53], and other tests could be conducted to scientifically asses the obtained results.

The formalin test for nociception, which is mainly used with rats and mice, involves moderate and continuous pain generated by injured tissue. This test in mice is a well-founded and trustworthy model of nociception and is sensitive for various classes of analgesic drugs [56]. Plantar injections of formalin in the mouse induce a characteristic behaviour evoked in two temporal phases. The first phase is observed immediately after the formalin injection, followed by a quiescent period, and then a second phase appears, which lasts until the end of the period of observation. The first phase depends upon direct stimulation of nociceptors, whereas the second involves a period of sensitization during which inflammatory phenomena occurs, involving the release of pain mediators such as histamine, serotonin, prostaglandins, bradykinin, and cytokines. Opioid analgesics appear to be antinociceptive for both phases, whereas NSAIDs seem to suppress only the second phase [53]. In our research, AG substantially diminished the second phase of the formalin test when administered 24 h before the test, thus confirming the results obtained in the writhing test.

Furthermore, in vivo tests of AG show the efficiency of the drug in terms of increase of nociceptive threshold after zymosan administration. Associated with paw edema, the injection of zymosan in the mouse paw produces a decrease of pain thresholds, which indicates a state of hyperalgesia [44,45]. In this pain model, a time-dependent hyperalgesia was noticed after zymosan administration and measured as a time-dependent reduction in the latency to respond to the thermal stimuli applied to the injected paw compared to the baseline measurements. We also have outstanding results in terms of the highest increase in pain threshold after AG administration. We observed that AG caused a significant reduction of zymosan-induced hyperalgesia until 48 h after administration.

It is well known that several components of liquorice can suppress COX-2 [57,58] and this seems to be true also for AG [16]. In addition to above-mentioned effects of AG on COX-2 [51,52], in skin tumor, AG significantly inhibited NF-κB, COX-2, prostaglandin E_2_ (PGE_2_), and nitric oxide (NO) levels [59]. Furthermore, in focal cerebral ischemic-reperfusion injury in mice, AG administered for seven days significantly improved neurofunction, decreased infarct size, and suppressed edema [60]. The neuroprotective effect of AG was associated with a significant reduction in IL-1, TNF-α, COX-2, iNOS, NF-κB, and GFAP levels [60]. Further recent studies showed that AG downregulates the expression of iNOS and COX-2 in the inflamed kidney [61] and skin [62], blocking NF-κB activation. In HaCaT cells, AG inhibited the UV-B-mediated increase in intracellular ROS and downregulated the release of IL-6, IL-1α, IL-1β, TNF-α, and PGE_2_ [63]. In LPS-stimulated mouse endometrial epithelial cells, AG considerably repressed LPS-induced TNF-α, IL-1β, NO, and PGE_2_ production, attenuated LPS-induced iNOS, COX-2, and TLR4 expression and NF-κB activation [64].

Our docking studies, for the first time, suggest that AG’s effects on inflammation and nociception might also depend upon the interaction with mPGES-1/2, COX-1/2, and 5-LO. In fact, from the analysis of the results, it emerged that AG interacted with key amino acids of mPGES-2 and COX-2, highlighting a preferential binding with these two isoforms. AG seems to locate better in the binding pocket of COX-2 as it interacts with key amino acids like Trp387, Ser530 (H-bonds), and Arg120 (salt bridge). Moreover, by combined rigid and flexible molecular docking studies, two possible mechanisms of interaction between AG and 5-LO were proposed: non-redox competitive binding and Fe^2+^ complexation. Here, the binding energy calculated is lower compared to those obtained with the other proteins, namely mPGES-1 and 2, COX-1 and 2 (data not shown), but consistent with putative inhibitor activity. These data suggest that further experiments should be performed to evaluate the effects of AG’s interaction with 5-LO.

## 4. Materials and Methods

### 4.1. Reagents

Ethylenediaminetetraacetate (EDTA), enhanced chemiluminescence detection kit (ECL), and glycyrrhizic acid ammonium salt from glycyrrhiza root (≥95%) were purchased from Sigma-Aldrich (Milan, Italy).The proteome profiler mouse cytokine array kit was from R&D System (Abingdon, UK). All the other reagents were from Carlo Erba (Milan, Italy), unless otherwise specified.

### 4.2. Animals and Ethical Statement

We used male CD-1 mice (Harlan, Italy) weighing 25 g in all of the experiments. Mice were housed in colony cages (seven mice per cage) under standard conditions of light, temperature, and relative humidity for at least one week before the start of experimental sessions. Food and water were available ad libitum.

All experiments were performed according to Legislative Decree 26/14, which implements the European Directive 2010/63/UE on laboratory animal protection in Italy, and were approved by the local ethics committee. Animal studies are reported in accordance with the ARRIVE (Animal Research: Reporting of In Vivo Experiments) guidelines [65,66]. The research protocol was approved by the Service for Biotechnology and Animal Welfare of the Istituto Superiore di Sanità and authorized by the Italian Ministry of Health.

### 4.3. Edema Induced by Zymosan

Edema was induced with a subcutaneous injection of 2.5% *w/v* zymosan A in saline, in the dorsal surface of the right hind paw (20 μL/paw). Paw volume was measured three times before the injections and at 1, 2, 3, 4, 24, and 48 h thereafter by the use of hydroplethysmometer specially modified for small volumes (Ugo Basile, Italy). AG (50 or 150 mg/kg) and vehicle (Hepes, 10 mL/kg) were administered intraperitoneally (i.p.) 10 min before zymosan A. The increase in paw volume was evaluated as percentage difference between the paw volume at each time point and the basal paw volume [67].

### 4.4. Writhing Test

Mice were given an intraperitoneal injection of 0.6% *v/v* acetic acid in a volume of 10 mL/kg. Acetic acid induces a series of writhes, consisting in abdominal contraction and hind limb extension [68]. AG (50 or 150 mg/kg) and vehicle (Hepes, 10 mL/kg) were administered i.p. 24 h before acetic acid. After vehicle or AG administration, the number of writhes were recorded over a 20 min period beginning 5 min after acetic acid injection.

### 4.5. Formalin Test

Male mice were split into three groups of ten animals each: vehicle (10 mL/kg, i.p.), AG (50 mg/kg, i.p.) and AG (150 mg/kg, i.p.). One percent formalin in saline was injected into the right mouse hind paw, twenty-four hours after saline and AG i.p. administrations. Formalin aroused nociceptive behavioural responses, such as licking and/or biting the injected paw, which are considered indices of nociception [69]. The nociceptive response showed a biphasic trend: an early phase (from 0 to 10 min after formalin injection) produced by the direct stimulation of peripheral nociceptors, and a late prolonged phase (from 15 to 40 min) which reflected the response to inflammatory pain. The total time the animal spent licking or biting its paw during the early and late phase of formalin-induced nociception was recorded. During the test, the mouse was located in a Plexiglas observation cage (30 × 14 × 12 cm) 1 h before the formalin administration to allow it to acclimatize to its surroundings.

### 4.6. Zymosan-Induced Hyperalgesia

In these experiments, AG (50 or 150 mg/kg) and vehicle (Hepes, 10 mL/kg) were administered intraperitoneally (i.p.) 10 min before a subcutaneous injection (20 μL/paw) of zymosan A (2.5% *w/v* in saline) into the dorsal surface of the right hind paw. Then, hyperalgesia measurements were performed [70]. The plantar test (Ugo Basile, Italy) was used to measure the sensitivity to a noxious heat stimulus with the aim of assessing thermal hyperalgesia after zymosan-induced inflammation of the mouse hind paw. The animals are located in cages with a glass floor covered with transparent plastic boxes and allowed to habituate to their surroundings for at least 1 h in a temperature-controlled room (21 °C) for three consecutive days prior to testing. On the test day, the animals were acclimatized to their environment for at least 1 h before paw withdrawal latency (PWL) was measured. Care was taken to initiate the test when the animal was at rest, not walking, with its hind paw in contact with the glass floor of the test apparatus. A radiant heat source was constantly directed at the mouse footpad until paw withdrawal, foot drumming, licking, or any other aversive action was observed. A timer started automatically when the heat source was activated, and a photocell stopped the timer when the mouse withdrew its hind paw. The heat source on the plantar apparatus was set to an intensity of 30 and a cut-off time of 15 s was used to avoid tissue damage. Animals were first tested to determine their baseline PWL; after zymosan injection, the PWL (seconds) of each animal in response to the plantar test was determined again at 1, 2, 3, 4, 5, 24, and 48 h.

### 4.7. Zymosan Peritonitis

Zymosan peritonitis was induced as previously reported [71,72]. Mice were injected intraperitoneally with 500 mg/kg of zymosan. Animals were killed by CO_2_ exposure, peritoneal cavities washed with 3 mL of PBS containing 3 mM EDTA, at different time points. Aliquots of the lavage fluids were then stained with Turk’s solution (0.01% crystal violet in 3% acetic acid) and differential counts performed using a Neubauer haemocytometer counting chamber (Thermo Fisher Scientific, Rome, Italy) and a light microscope. Lavage fluids were collected and used to measure the relative expression levels of ~40 cytokines and chemokines using the Mouse Cytokine Array Panel A from R&D System.

### 4.8. Cytokines and Chemokines Protein Array

Equal volumes (1.5 mL) of the peritoneal inflammatory fluids obtained at different time points (4 and 24 h) after the treatment with zymosan (500 mg/kg) or zymosan plus AG (150 mg/kg) were incubated with the precoated proteome profiler array membranes according to the manufacturer’s instructions. Dot plots were detected by using ECL detection kit and ImageQuant 400 GE Healthcare (GE Healthcare Europe GmbH, Milan, Italy) and successively quantified as optical density (INT/mm_2_) using GS 800 imaging densitometer software (Bio-Rad Laboratories, Milan, Italy) as previously described [73].

### 4.9. Molecular Docking Input Files Preparation

The crystal structure of mPGES1 in complex with inhibitors (5TL9) [74], mPGES-2 in complex with indomethacin (1Z9H) [24], COX-1 in complex with mofezolac (5WBE) [25], COX-2 in complex with meclofenamic acid (5IKQ) [38], and 5-lipoxygenase (3V99) [32] were used as molecular targets for the docking studies. All the protein 3D structures were produced using the Schrödinger Protein Preparation Wizard workflow [75]. First, all missing hydrogen atoms were added, bond orders were properly assigned, partial charges calculated, water molecules were eliminated, and protein termini were capped. Glycyrrhizic acid was built with Maestro’s Build Panel (Maestro version 10.2, 2015), processed with LigPrep (LigPrep version 3.4, 2015) and, finally, minimized with the OPLS 2005 force field [76].

### 4.10. Docking Experiments

The cocrystallized inhibitors were used to generate the grid necessary for the molecular docking experiments, using the following coordinates: 10.75 (x), 15.07 (y), 28.45 (z) for mPGES-1; −21.10 (x), 54.51 (y), 11.75 (z) for mPGES-2; 19.34 (x), 2.67 (y), 31.22 (z) for COX-1; 24.25 (x), 3.52 (y), 33.07 (z) for COX-2; 9.21 (x), −82.00 (y), −32.69 (z) for 5-LO. The boundaries of the inner box were extended of 10 Å for COX-1 and mPGES1 and 13 Å for mPGES-2 and COX-2 in the three directions of space. Regarding 5-LO, in both calculations, the centre of the grid was set close to Asn544 and, in the rigid docking approach, the outer box boundaries were set at 40 Å from it. For the in silico docking experiments, the software Glide [77,78,79] was used. In the rigid docking approach, in a preliminary phase, in standard precision (SP) mode, 10,000 ligand poses were kept, and the best 800 poses were selected for energy minimization. Once the minimization was completed, only one output structure was saved for each ligand. After this phase, a post-docking optimization of the obtained molecules was carried out, setting the rejection cut-off at 0.5 kcal/mol and a maximum of ten poses per ligand were kept. The poses generated in the initial phase, were submitted to a second optimizations phase in extra precision glide (XP) mode. In this last phase, the post-docking optimization of the docking poses was performed accounting for a maximum of ten poses with the same parameters for the selection of initial poses and rejection cut off used during the SP docking experiments. The induced fit docking was carried out using the same settings of the rigid docking XP mode, but allowing a certain degree of flexibility to amino acids of the catalytic pocket. The first step consists in docking the ligand in the binding site with Glide [77,78,79]; then, the protein side chains are re-oriented with Prime [80,81] with the aim of better accommodating the molecule. At the last stage, a second rigid docking inside the new cavity is carried out.

### 4.11. Data Analysis and Statistics

The results achieved are given as the mean ± SEM. Statistical analysis was carried out by using one-way ANOVA followed by Dunnett’s post-test when comparing more than two groups. In certain cases, one sample *t*-test was used to evaluate significance against the hypothetical zero value. Statistical analysis was conducted by using GraphPad Prism 6.0 software (San Diego, CA, USA). Data were considered statistically significant when a value of *p* < 0.05 was achieved. The data and statistical analysis comply with the recommendations on experimental design and analysis [82].

## 5. Conclusions

Results of the present study indicated that AG possesses long-lasting anti-inflammatory and antinociceptive effects as observed 24–48 h after the administration. Our data also suggest that its anti-inflammatory and antinociceptive effects might be attributed to the inhibition of the levels of different pro-inflammatory cytokines and chemokines. Taken together, all these findings indicate that AG is a long-acting therapeutic agent for the treatment of painful conditions and inflammatory-related diseases.

## Figures and Tables

**Figure 1 molecules-24-02453-f001:**
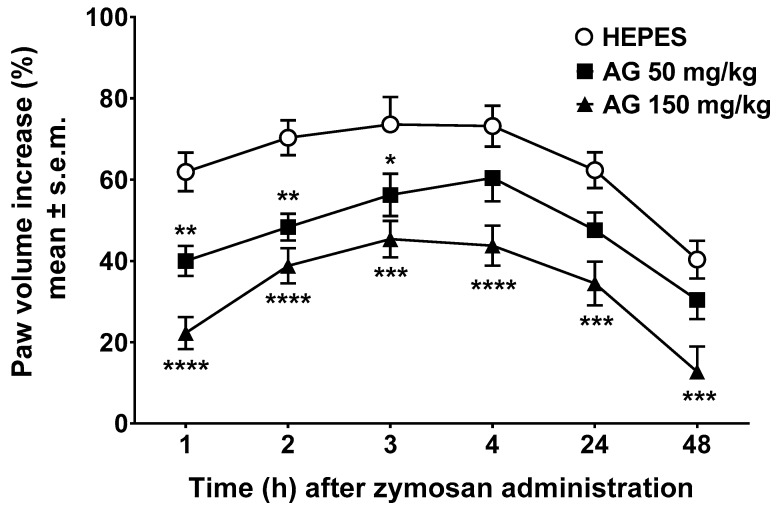
Zymosan-induced paw edema. Effects induced by vehicle (Hepes, 10 mL/kg, intraperitoneally (i.p.)) and ammonium glycyrrhizate (AG, 50 or 150 mg/kg, i.p.) administered 10 min before zymosan (2.5% *w/v* in saline, 20 µL/paw). * denotes *p* < 0.05, ** denotes *p* < 0.01, *** denotes *p* < 0.001 and **** denotes *p* < 0.0001 vs. Vehicle. *n* = 12.

**Figure 2 molecules-24-02453-f002:**
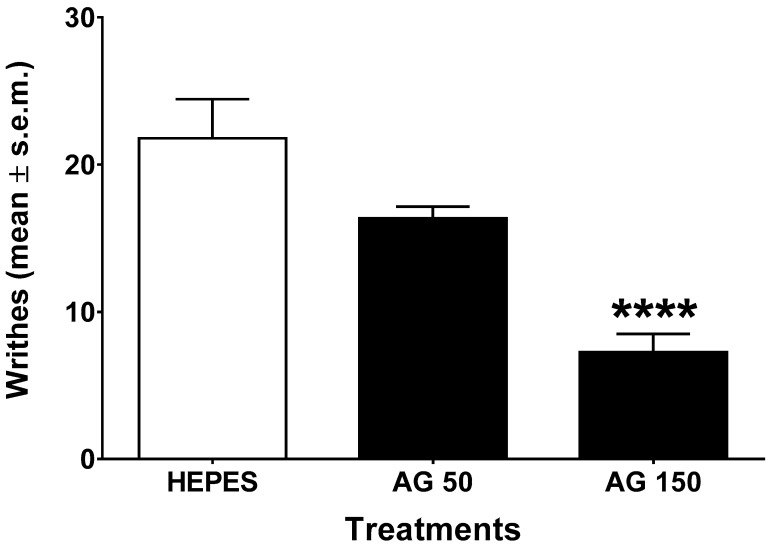
Writhing test. Effects induced by vehicle (Hepes, 10 mL/kg, intraperitoneally (i.p.)) and ammonium glycyrrhizate (AG, 50 or 150 mg/kg, i.p.) administered 24 h before acetic acid (0.6% *v/v* in salina, 10 mL/kg, i.p.). **** denotes *p* < 0.0001 vs. Vehicle. *n* = 9.

**Figure 3 molecules-24-02453-f003:**
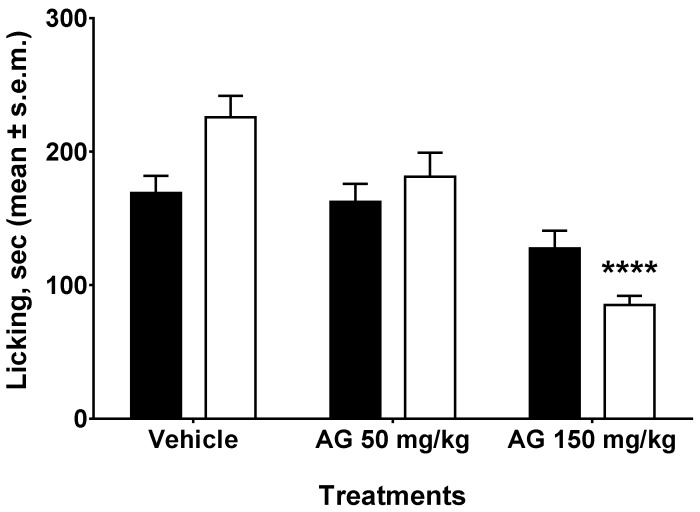
Formalin test. Effects induced by vehicle (Hepes, 10 mL/kg, i.p.) and ammonium glycyrrhizate (AG, 50 or 150 mg/kg, i.p.) administered 24 h before formalin (1% in saline, 20 µL/paw) in the formalin test. Black bars represent the early phase and the white bars represent the late phase of the formalin test. **** is for *p* < 0.0001 vs. Vehicle. *n* = 10.

**Figure 4 molecules-24-02453-f004:**
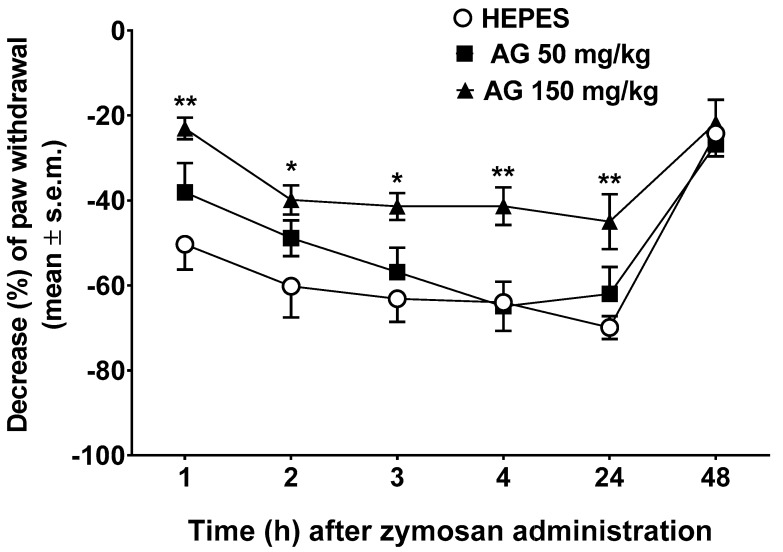
Zymosan-induced hyperalgesia. Effects induced by vehicle (Hepes, 10 mL/kg, intraperitoneally (i.p.)) and ammonium glycyrrhizate (AG, 50 or 150 mg/kg, i.p.) administered 10 min before zymosan (2.5% *w/v* in saline, 20 µL/paw). * denotes *p* < 0.05 and ** denotes *p* < 0.01 vs. Vehicle. *n* = 10.

**Figure 5 molecules-24-02453-f005:**
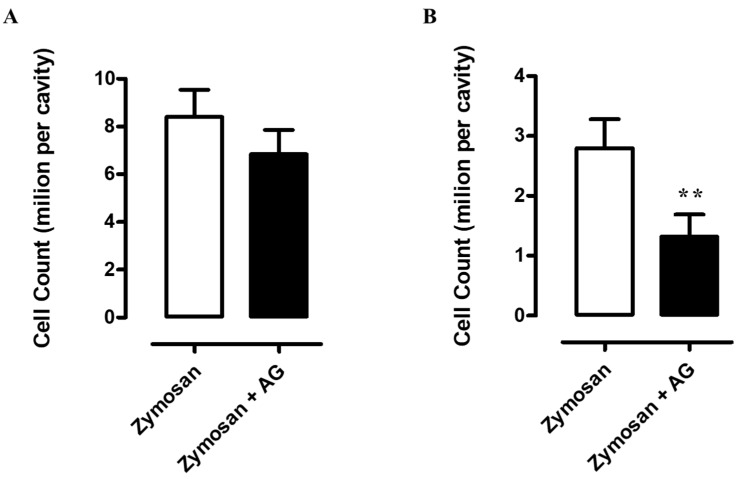
(**A**) Cell infiltration into mouse peritoneal cavity of mice that have received an intraperitoneal (i.p.) injection of zymosan (500 mg/kg) or zymosan + AG (150 mg/kg) at 4 and (**B**) 24 h. Bars represent group mean values ± Standard Error of the Mean (S.E.M) of three separate experiments with *n* = 7 mice. ** *p* < 0.01 vs. zymosan-treated mice.

**Figure 6 molecules-24-02453-f006:**
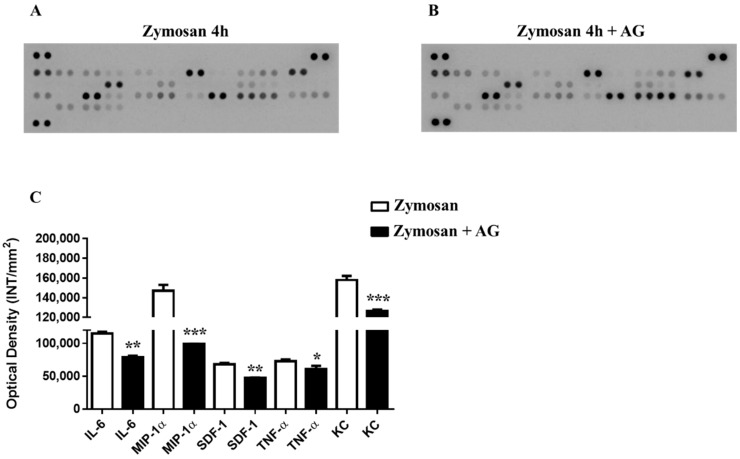
(**A**) Inflammatory fluids in zymosan and zymosan + AG-injected mice at 4 h. Inflammatory fluids obtained from peritoneal cavities were assayed using a proteome profiler cytokine array. Mean changes (± S.E.M) between zymosan (500 mg/kg) and (**B**) zymosan + AG-injected mice (150 mg/kg) of three separate experiments with *n* = 7 mice, (**C**) are expressed as increase in the optical density (INT/mm_2_) between the two groups. * *p* < 0.05, ** *p* < 0.01 and *** *p* < 0.001 vs. zymosan-treated mice.

**Figure 7 molecules-24-02453-f007:**
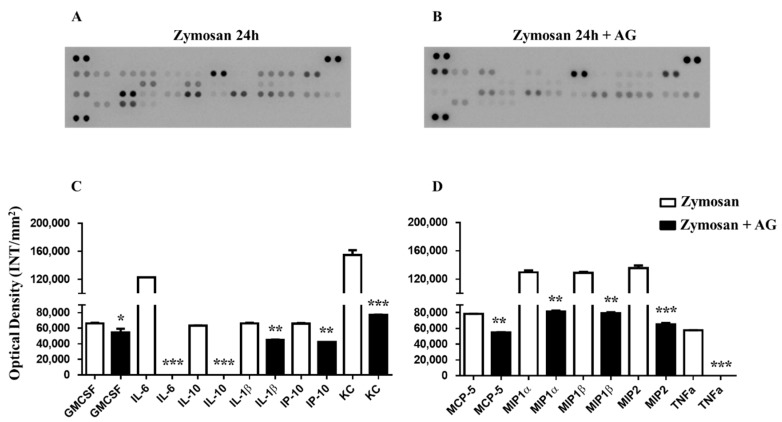
(**A**) Inflammatory fluids in zymosan and zymosan + AG-injected mice at 24 h. Inflammatory fluids obtained from peritoneal cavities were assayed using a proteome profiler cytokine array. (**B**) Mean changes (± S.E.M) between zymosan (500 mg/kg) and zymosan + AG-injected mice (150 mg/kg) of three separate experiments with *n* = 7 mice, (**C, D**) are expressed as increase in the optical density (INT/mm_2_) between the two groups. * *p* < 0.05, ** *p* < 0.01 and *** *p* < 0.001 vs. zymosan-treated mice.

**Figure 8 molecules-24-02453-f008:**
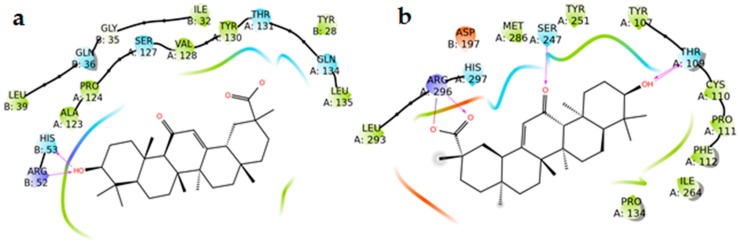
(**a**) 2D panels represent the interactions between AG with the microsomal prostaglandin E synthase 1 (mPGES-1) and 2 (mPGES-2) (**b**) binding sites. Positively charged residues are colored violet, negatively charged residues are colored red, polar residues are colored light blue, hydrophobic residues are colored green. Red-to-blue lines represent salt bridges and H-bonds (side chain) are reported as dotted pink arrows.

**Figure 9 molecules-24-02453-f009:**
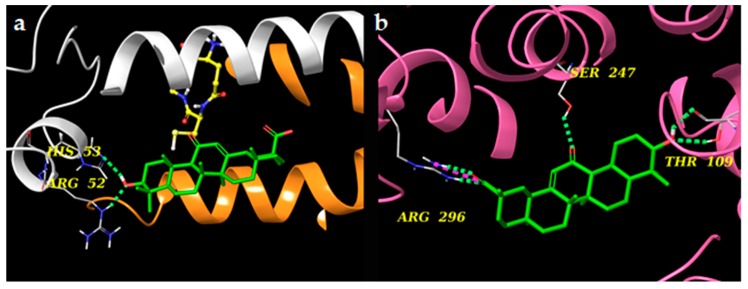
(**a**) 3D models of AG (colored by atom types: C green, O red, polar H white) in the binding site of microsomal prostaglandin E synthase 1 (mPGES-1) (panel a, chain A and chain B are reported in orange and white ribbons, respectively) and (**b**) microsomal prostaglandin E synthase 2 (mPGES-2) (chain A is reported in pink ribbons). Green dotted lines represent H-bonds and pink dotted lines represent salt bridges.

**Figure 10 molecules-24-02453-f010:**
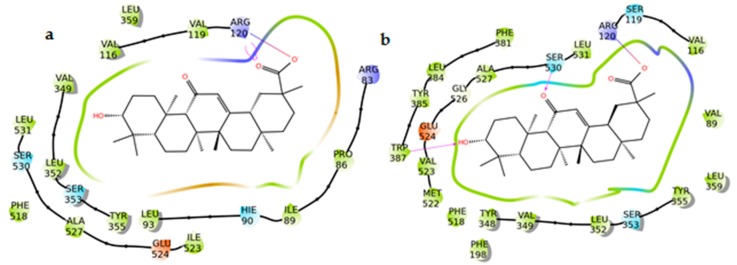
(**a**) 2D panels represent the interactions between AG with cyclooxygenase 1 (COX-1) and (**b**) cyclooxygenase 2 (COX-2) binding sites. Positively charged residues are colored violet, negatively charged residues are colored red, polar residues are colored light blue, hydrophobic residues are colored green. Red-to-blue lines represent salt bridges and H-bonds (side chain) are reported as dotted pink arrows. Neutral histidine with hydrogen on the δ nitrogen is shown as HIE.

**Figure 11 molecules-24-02453-f011:**
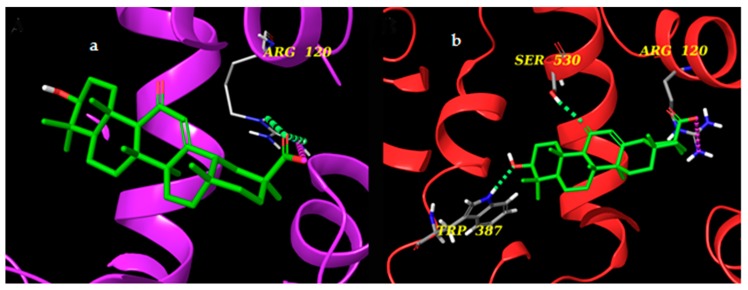
(**a**) 3D models of AG (colored by atom types: C green, O red, polar H white) in the binding site of cyclooxygenase 1 (COX-1) (purple ribbons) and (**b**) cyclooxygenase 2 (COX-2) (red ribbons). Green dotted lines represent H-bonds and pink dotted lines represent salt bridges.

**Figure 12 molecules-24-02453-f012:**
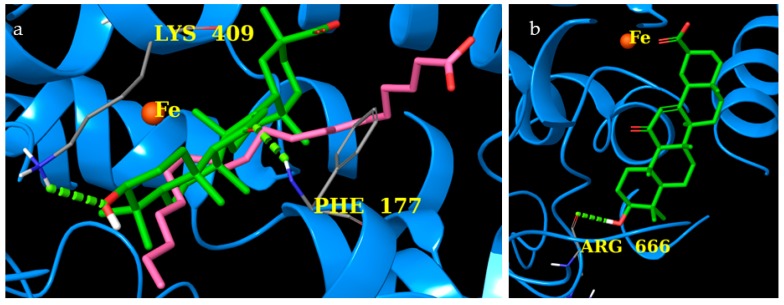
(**a**) Rigid docking and induced fit (**b**) of AG (colored by atom types: C green, O red, polar H white) in the binding site of 5-lipoxygenase (5-LO) (blue ribbons and Fe^2+^ as an orange sphere). Green dotted lines represent H-bonds, and original arachidonic acid (AA) molecule is in pink.

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
