# Peer review of "Long-Lasting Anti-Inflammatory and Antinociceptive Effects of Acute Ammonium Glycyrrhizinate Administration: Pharmacological, Biochemical, and Docking Studies"

_molecules, 2019, doi:10.3390/molecules24132453_

Round 1
Reviewer 1 Report
In this study the authors have evaluated the long-lasting effects induced by Ammonium Glycyrrhizinate (AG) after a single administration in mice using animal models of pain and inflammation together with biochemical and docking studies. A single intraperitoneal injection of AG was able to produce anti-inflammatory effects in zymosan-induced paw edema and in zymosan-induced peritonitis. In several animal models of pain as the writhing test, the formalin test and the hyperalgesia induced by zymosan, AG administered 24h before the tests was able to induce a strong antinociceptive effect. Molecular docking studies revealed that AG could have a higher affinity for microsomal prostaglandin E synthase type-2 with respect to type-1 whereas AG seems to locate better in the binding pocket of cyclooxygenase (COX) -2 compared to COX-1.
Specific Comments:
This is technically well performed study but the authors need to address several missing links before it can be even considered for publication. Specific points that the authors need to address are as follows:
The molecular mechanism(s) by which AG exhibits its anti-inflammatory effects are not clear? For example, whether deletion of COX-2 by siRNA abrogates the observed anti-inflammatory effects of AG should be analyzed?
Whether AG can also affect COX-2 activity/expression should be analyzed.
Whether AG can effect IL-1, TNF-α, iNOS,and NF-κB levels in zymogen-induced paw edema model should be analyzed.
Acute toxicity studies should be performed to establish the safety of AG.
Author Response
Reviewer 1
Specific Comments:
This is technically well performed study but the authors need to address several missing links before it can be even considered for publication. Specific points that the authors need to address are as follows:
The molecular mechanism(s) by which AG exhibits its anti-inflammatory effects are not clear? For example, whether deletion of COX-2 by siRNA abrogates the observed anti-inflammatory effects of AG should be analyzed?
We thanks the Reviewer for the helpful comment. COX-2 is an enzyme playing key roles in the mechanism of pain and inflammation, induced only during inflammatory processes and probably reflects no role in the safe tissue. Thus, selective COX-2 inhibition can significantly reduce the adverse effects of traditional NSAIDs like aspirin, ibuprofen, etc.. RNAi technology has been successfully used to silence COX-2 protein in different in vitro models. The use of innovative RNAi-based techniques has enabled researchers to better study the molecular and phenotypical loss of function of COX-2 gene by performing experiments based on a strong COX-2 silencing. Accordingly to the aim of our study, we induced inflammation and pain using different in vivo approaches in order to investigate AG effects on these parameters, thus we need plenty COX-2 expression and function.
Whether AG can also affect COX-2 activity/expression should be analyzed.
We thank the Reviewer for the helpful comment. A lot of data demonstrated that AG is able to modify COX-2 activity/expression (for a recent review on the AG effects on COX-2 in rheumatoid arthritis, please see Huang et al., Oncotarget. 7(2):1193-202, 2016, doi: 10.18632/oncotarget.6200) and we reported some further evidences in the Discussion section of the manuscript (please see, page 11 lines 322-333). In addition to these data, for the first time we suggest that AG effects on inflammation and nociception might also depend upon the direct inhibitory interaction with the active site of mPGES-2 and COX-2 and not only through the modulation of their expression. The observations provide a mechanism for the already known inhibitory effects of AG on PGE2 synthesis.
Whether AG can effect IL-1, TNF-α, iNOS, and NF-κB levels in zymogen-induced paw edema model should be analyzed.
We thank the Reviewer for the helpful comment. In the first series of experiments, we have tried to study the IL-1b and TNF-a level in the exudate from the paw of zymosan-treated animals, obtaining inconsistent result probably due to low volume samples. Thus, we decided to perform zymosan peritonitis and cytokines-chemokines protein array, in order to obtain better results from our experiments and investigate the relative expression levels of ~ 40 cytokines and chemokines, many of these never measured before. We did not perform experiments on iNOS and NF-kB level, since previous studies already demonstrated that AG is able to modify both iNOS and NF-kB levels in several models of inflammation (please see, Liu et al., Medchemcomm. 2018 Jul 19;9(9):1502-1510. doi: 10.1039/c8md00288f; Wang et al., Microb Pathog. 2017 Aug;109:110-113. doi: 10.1016/j.micpath.2017.05.032).
Acute toxicity studies should be performed to establish the safety of AG.
We thank the Reviewer for the helpful comment. In the revised version of the manuscript, we add some information on the AG safety and toxicity. In the revised version of the manuscript, please see page 10, lines 272-273.
Reviewer 2 Report
The manuscript is interesting as it explores the mechanisms associated with the analgesic and anti-inflammatory effects of ammonium glycyrrhizinate. The anti-nociceptive and anti-inflammatory actions of the compound have been previously shown by the same group including in the zymosan-induced oedema and formalin-induced nociception models. However, this new study approached the mechanisms involved in such effects.
It was unclear why did the authors chose to perform a docking analysis for COX/PG molecules only and not for other pathways involved in inflammation and pain.
The authors did not cite their own previous results with this same compound, and ignored similarities between the present study and the one published in 2014 ( doi: 10.2147/IJN.S55066).
Consideting that the manuscript´s idea is not novel in regards of ammonium glycyrrhizinate, but it brings interesting information on the mechanisms of action of this compound, this referee considers that such mechanisms need to be further discussed in the light of previous studies and also further evaluated in terms of pharmacological targets in the docking analysis.
Author Response
Reviewer 2
The manuscript is interesting as it explores the mechanisms associated with the analgesic and anti-inflammatory effects of ammonium glycyrrhizinate. The anti-nociceptive and anti-inflammatory actions of the compound have been previously shown by the same group including in the zymosan-induced oedema and formalin-induced nociception models. However, this new study approached the mechanisms involved in such effects.
It was unclear why did the authors chose to perform a docking analysis for COX/PG molecules only and not for other pathways involved in inflammation and pain.
We thank the Reviewer for the helpful comment. Several data indicate that AG significantly inhibited COX-2/PGE2 levels, thus we decided to investigate AG docking profile using the mPGES-1 and mPGES-2 substrates and also COX-1 and COX-2, key enzymes involved in the eicosanoid metabolism pathway. However, we performed further experiments on AG docking using LOX, and the results are added in the revised version of the manuscript. Please see page 9-10, lines 234-260 in the revised version of the manuscript.
The authors did not cite their own previous results with this same compound, and ignored similarities between the present study and the one published in 2014 (doi: 10.2147/IJN.S55066).
We thank the Reviewer for the helpful comment. As suggested, in the revised version of the manuscript we report our previous data on AG. Please see page 10-11, lines 285-288 in the revised version of the manuscript.
Consideting that the manuscript´s idea is not novel in regards of ammonium glycyrrhizinate, but it brings interesting information on the mechanisms of action of this compound, this referee considers that such mechanisms need to be further discussed in the light of previous studies and also further evaluated in terms of pharmacological targets in the docking analysis.
We thank the Reviewer for the helpful comment. As suggested, we improve the manuscript, discussing further new data on LOX docking studies and AG effects on those cytokines-chemokines never investigated before. To this aim, new materials and methods and results have been added to the revised version of the manuscript. Please also see the Discussion section, where these new data are speculated in relation to those of COX-2/mPGEs pathway.
Round 2
Reviewer 1 Report
The authors have addressed all my concerns.
Author Response
We thanks the Reviewer for the helpful comment and suggestions.
Reviewer 2 Report
Somehow, grammar has become poorer following revision.
Considering that the manuscript´s idea is not novel in regards of ammonium glycyrrhizinate, but it brings interesting information on the mechanisms of action of this compound, this referee considers that such mechanisms need to be further discussed in the light of previous studies and also further evaluated in terms of pharmacological targets in the docking analysis.
Author Response
Reviewer 2
Somehow, grammar has become poorer following revision.
We thank the Reviewer for the helpful suggestion. In the second version of the manuscript, we have improved English style and we corrected grammar mistakes.
Considering that the manuscript´s idea is not novel in regards of ammonium glycyrrhizinate, but it brings interesting information on the mechanisms of action of this compound, this referee considers that such mechanisms need to be further discussed in the light of previous studies and also further evaluated in terms of pharmacological targets in the docking analysis.
We thank the Reviewer for the helpful suggestion. As you suggested, in the second version of the revised manuscript we performed further molecular docking experiment on the 5-lipoxygenase–activating protein (FLAP), known for activating the conversion of arachidonic acid into 5-(S)-hydroperoxy-6,8,11,14-eicosatetraenoic acid (5-HpETE) and leukotriene A4 (LTA4). The crystal structure of the human FLAP bound to a known inhibitor (PDB: 2Q7M) was used during computational studies by software Glide. The molecular docking of both MK-591 and AG was carried out and comparing the binding mode of AG with respect to the co-crystallized inhibitor, MK-591 forms several interactions with Lys116 and a π- π interaction with Tyr112 essential for the inhibitory activity. On the other hand, AG was not able to interact with these key amino acids; the only bonds formed are with Lys29, far from the binding cavity. The results suggested a non-optimal binding between AG and FLAP.